# Assessing Microstructure Tensile Properties Relationships in Al-7Si-Mg Alloys via Multiple Regression

**Cássio Silva** [1], **André Barros** [1], **Talita Vida** [2], **Amauri Garcia** [1], **Noé Cheung** [1], **Danieli A. P. Reis** [3] **and Crystopher Brito** [2],*

1. Department of Manufacturing and Materials Engineering, University of Campinas, UNICAMP, Campinas 13083-860, Brazil; cassio.augusto92@gmail.com (C.S.); andre.barros13@hotmail.com (A.B.); amaurig@fem.unicamp.br (A.G.); cheung@fem.unicamp.br (N.C.)
2. Faculty of Engineering of São João da Boa Vista, São Paulo State University, UNESP, São João da Boa Vista 13876-750, Brazil; talita.vida@unesp.br
3. Institute of Science and Technology, Federal University of São Paulo, UNIFESP, São José dos Campos 12231-280, Brazil; danieli.reis@unifesp.br
* Correspondence: crystopher.brito@unesp.br; Tel.: +55-19-3638-2444

**Abstract:** The development of Al-based alloys presumes a detailed understanding of the microstructure evolution during solidification since the as-solidified microstructure also has effects on the subsequent thermo-mechanical processing. In the present investigation Al-7wt.%Si-xMg (x = 0.5 and 1 wt.%) alloys are subjected to transient directional solidification with a view to characterizing the microstructure evolution, with special focus on both dendritic evolution and the inherent features of the $Mg_2Si$ and $\pi$-AlSiFeMg intermetallics. Experimental power-type functions relating the primary, secondary and tertiary interdendritic spacings to the solidification cooling rate and growth rate are developed. It is observed that the Mg content added to the Al-7wt.%Si alloy and the consequent increase in the $Mg_2Si$ fraction tends to increase the values of the primary dendritic spacing. However, this same behavior is not verified for the growth evolution of dendritic side branches. A multiple linear regression (MLR) analysis is developed permitting quantitative correlations for the prediction of tensile properties and hardness from microstructural parameters to be established. The increase in the Mg alloy content from 0.5 to 1 was shown to promote an increase in both the ultimate tensile strength ($\sigma_u$) and elongation.

**Keywords:** Al-Si-Mg alloys; solidification; microstructure; mechanical properties; regression analysis

## 1. Introduction

Due to their unique combinations of properties, such as a high strength-to-weight ratio associated with suitable corrosion resistance and excellent thermal conductivity, Al-based alloys are materials of prime importance in many manufacturing technologies [1]. Alloying elements are commonly added to these alloys to improve the alloy final properties either through changes in the morphology of the microstructural phases or through the formation of new ones. To classify such elements, they can be divided into three main groups: basic, ancillary addition and impurity [2]. Si is a classic example of a basic element present in several Al-based alloy castings, and its content mostly lies between 5 and 12 wt.%. A related point to consider is that, depending on the final application, the same element can play different roles, thus belonging to more than one of these groups. This is the case for Mg, which is regarded as a basic element in the 5xxx series alloys (Al-Mg system) [3] but as an ancillary in the commercial A356 and A357 series [4]. The development of Al-based alloys with desirable properties, however, is much more complex than simply choosing solute contents, as demonstrated in previous works [5–8]. It also presumes a detailed understanding of the microstructure evolution during solidification since, besides

governing the final properties of as-solidified parts, the solidification microstructure also has effects on the subsequent thermo-mechanical processing [9].

Many attributes of Al-Si-Mg alloys, such as excellent fluidity, exceptional formability and proper corrosion resistance, render them attractive to the manufacturing of aerospace and automotive components [10]. Just as what happens with the binary Al-Si system, the presence of Si in these alloys is one of the main reasons for their low coefficient of thermal expansion and increased wear resistance [11]. Mg is present not only to guarantee improved mechanical properties in the as-cast state but also to enable these alloys to be heat-treatable through the precipitation of $Mg_2Si$ intermetallic compounds (IMCs), which are determinant to the increase in hardness [11–13]. Overall, the final properties of any part made of such alloys will be governed by microstructure features. That is why it is so useful and important to understand the relationship between their microstructure and mechanical properties. In as-cast Al-Si-Mg alloys, for example, the microstructural morphology of the primary Al-rich phase is usually dendritic [14]. From a quantitative metallography perspective, the measurement of the spacing between the Al-rich dendritic arms is an efficient strategy to analyze the overall fineness of the cast. The dendritic growth is indeed a key microstructure feature that can be quantitatively represented by the measurements of primary ($\lambda_1$), secondary ($\lambda_2$) and tertiary ($\lambda_3$) dendritic arm spacings [15]. However, the use of such microstructural spacings in approaches to predicting the mechanical properties of Al-Si-Mg alloys has not been fully explored.

Previous works developed equations for predicting the mechanical response of different metallic alloys through simple linear regression (SLR). For instance, Brito et al. [16] proposed experimental Hall-Petch equations to represent the relationships between microhardness and dendritic arm spacings in Al-3wt.%Mg (-1wt.%Si) alloys. For a ternary Sn-Zn-Cu alloy, Silva et al. [17] derived a Hall-Petch expression relating the hardness to the eutectic interphase spacing. Kakitani et al. [18] also proposed experimental equations correlating tensile properties with the eutectic colony spacing for an Al-Ni eutectic alloy. In the work of Chen et al. [10], the relationships between the tensile properties and $\lambda_2$ for Al-7wt.%Si-Mg alloys were evaluated using equations in the form of: $Y = a \cdot \lambda_2{}^2 + b \cdot \lambda_2 + c$, where Y represented the yield strength ($\sigma_{YS}$), the ultimate strength ($\sigma_{UTS}$) and the elongation to fracture (E%), and a, b and c were coefficients. However, models based on multiple regression (MR) analysis can be more adequate. Schon et al. [19], for example, noticed that multiple linear regression (MLR) models provide more consistent results in evaluating the ultimate tensile strength of SAC207 and SAC307 alloys, as compared with SLR. Kwak et al. [20] have proposed an MLR algorithm to predict the mechanical properties of directionally solidified Ti-Al alloys, and all of the coefficient of determination ($R^2$) values were 0.6 or above, meaning they had an effective $R^2$ value. Suh et al. [21] used MR models (MLR—multiple linear regression and MPR—multiple polynomial regression) to develop quantitative correlations between yield strength and the microstructural and textural variables of AZ31 sheets, but both models tended to overestimate the yield strength when compared with the measured values.

Besides the above-mentioned aspects, the fraction, morphology, size and distribution of the other phases forming the microstructure of Al-Si-Mg alloys are also decisive for the resulting mechanical behavior. $Mg_2Si$ IMCs are known to play a key role, since the rise in their faction is expected to promote significant improvements in the tensile strength [10]. Analyzing Al-7Si-(2–2.5) Mg alloys, Li et al. [22] observed that the precipitation of coarse Chinese script $Mg_2Si$ particles indeed increases the ultimate tensile strength but decreases the ductility. It is also worth mentioning that Fe is a common impurity that, even in residual traces, is capable of forming IMCs such as β-AlFeSi and π-AlSiFeMg phases, which are known to promote embrittlement and to decrease the corrosion resistance [23]. Analyzing the effects of Sr addition on the microstructure evolution of A356 and A357 alloys, Wang et al. [14] observed that the ductility decreases with increasing Mg content due to: (1) the increase in the size of π-AlSiFeMg IMCs and (2) the strengthening of the Al-rich matrix. Such microstructural features are dependent on an important parameter: the solidification

cooling rate [10,14,23,24]. It has been proven that this thermal parameter has a remarkable effect on the morphology of Si particles, as shown by Hearn et al. [25] for an Al-10wt.%Si alloy. Furthermore, analyzing directionally solidified Al-Si-Fe(-Ni) alloys, Canté et al. [26] observed that the fraction and circularity values of β-AlFeSi IMCs are reduced with the decrease in cooling rate.

A fascinating challenge is how to obtain experimental datasets for MLR analyses in order to develop quantitative correlations among solidification thermal parameters, microstructure evolution and mechanical properties for Al-Si-Mg alloys, considering cooling conditions similar to those of industrial processes. An effective answer consists in the use of a transient directional solidification technique—employed in such way as to generate a large spectrum of thermal histories along the length of the casting. This means the achievement of a wide range of a microstructural length scale at only one experimental run. Such approach is adopted in this work to produce Al-7wt.%Si-xMg (x = 0.5 and 1 wt.%) alloy samples that will be subjected to a comprehensive characterization. In summary, the main contributions of this paper are as follows:

- Proposal of experimental equations to represent the coarsening of dendrite arms of Al-7wt.%Si-xMg (x = 0.5 and 1 wt.%) alloys during solidification under transient heat flow conditions.
- A better understanding of the microstructure evolution, with a special focus on the inherent features of the $Mg_2Si$ and AlSiFeMg IMCs.
- Implementation of MLR analysis to determine the quantitative correlations for the prediction of tensile properties and hardness from microstructural parameters.

## 2. Materials and Methods

### 2.1. Al-Si-Mg Alloys

The investigated alloys were prepared from commercially pure Al, Si and Mg-Al master alloys. Table 1 presents the chemical composition of these metals. From now on, all alloying contents will be treated in terms of weight percent (wt.%). For example, the studied alloys will be written as Al-7Si-0.5Mg and Al-7Si-1Mg. Each alloy was directionally solidified (DS) in a solidification apparatus (Figure 1), in which the heat was removed from a water-cooled bottom, promoting vertical upward unsteady-state solidification. Details concerning the solidification apparatus can be found in a previous study [27]. The molten alloy was shed into a stainless-steel mold which was previously placed inside the casting chamber of the directional solidification device. The heating system was switched off when a melt superheating temperature that was about 20% higher than the alloy *liquidus* temperature ($T_L$) was achieved, and then the cooling system was activated, allowing for the start of the solidification. The cylindrical mold was instrumented with eight type K thermocouples (0.2 mm diameter wire) placed in the geometrical center of the cavity at different positions along its length, with a view to determining the cooling rate during solidification. The data acquisition rate was five points per second. Figure 1c shows the partial Al-7Si-xMg pseudo binary phase diagram, as calculated by the Thermo-Calc software (TTAL5 database) (Thermo-Calc 2018, Thermo-Calc software, Sweden), where vertical dashed lines indicate the nominal composition of each examined alloy.

**Table 1.** Chemical compositions (wt.%) of metals used to prepare the studied Al-Si-Mg alloys.

| Metals | Al | Si | Mg | Cu | Fe | Pb | Zn |
|---|---|---|---|---|---|---|---|
| Al | Balance | 0.006 | 0.0011 | 0.01 | 0.073 | 0.006 | 0.005 |
| Si | 0.109 | Balance | - | - | 0.316 | - | - |
| Mg-Al | 6.981 | 0.246 | Balance | 0.115 | 0.001 | - | 0.631 |

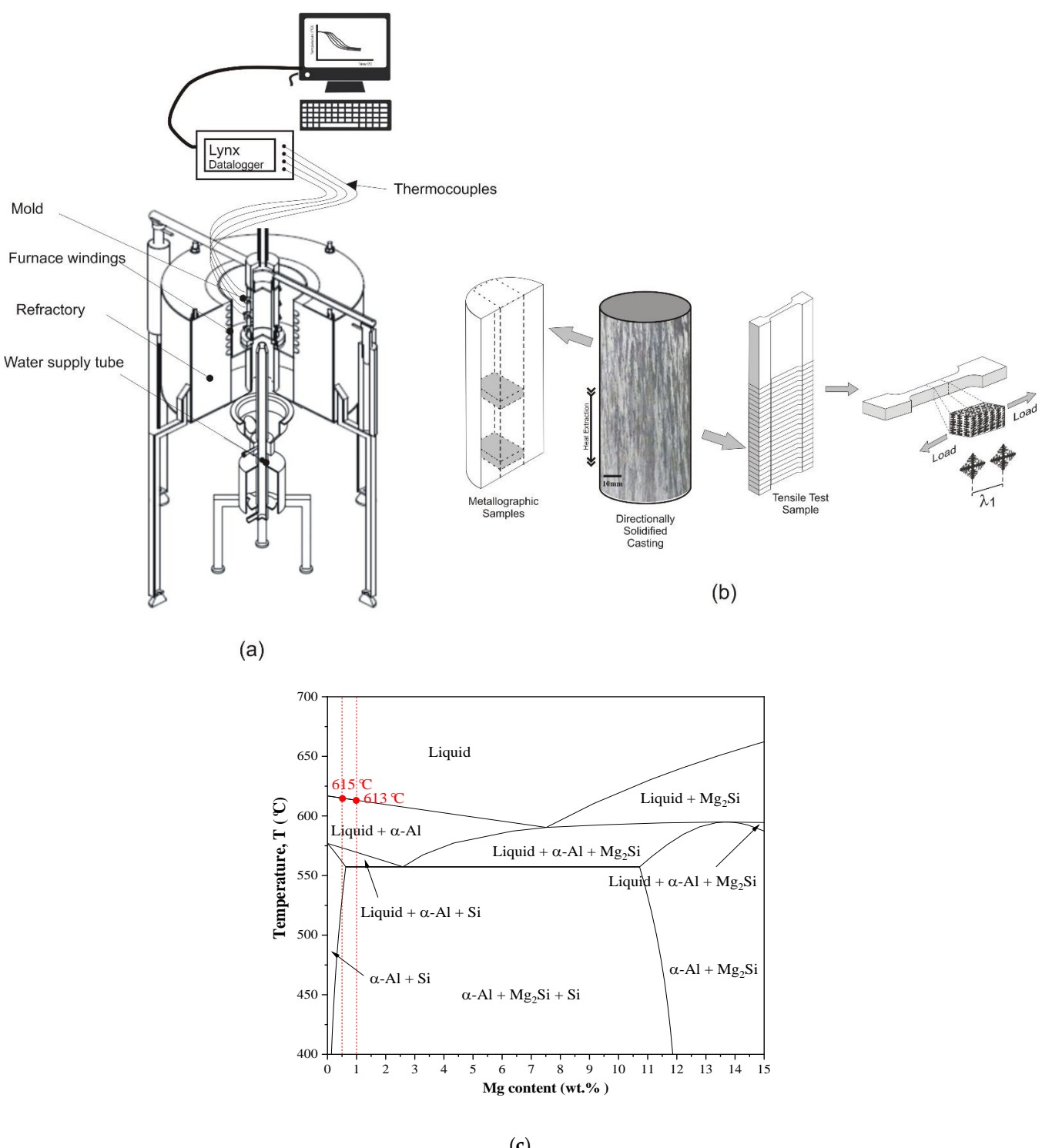

**Figure 1.** Schematic representations: (**a**) of the water-cooled upward directional solidification system and (**b**) of the procedures for the extraction of the samples for metallography and tensile tests along the length of the directionally solidified (DS) Al-7Si-xMg alloy castings; (**c**) Partial Al-7Si-xMg pseudo binary phase diagram calculated by the Thermo-Calc software (TTAL database), with vertical dashed lines indicating the nominal composition of each examined alloy.

The temperature–time data recorded during the solidification were used to determine the growth rate ($V_L$) through experimental techniques based on first time-derivative of the function $P = f(t)$, in which $P$ represents the position of each thermocouple from the mold's cooled surface and $t$ is the time corresponding to the *liquidus* isotherm passage by

each thermocouple. The profiles of the cooling rate ($\dot{T}_L$) were calculated using the plots of power law functions that represented the general tendency of the temperature–time data in a comprehensive region around the *liquidus* temperature. A fitted regression line was used only if its coefficient of determination ($R^2$) was higher than 0.9. More details of such procedures can also be found in previous studies [28,29].

### 2.2. Metallography

In order to reveal the solidification macrostructure, each DS casting was sectioned along its vertical axis, grounded and etched with a solution of 50% HCl + 50% $HNO_3$ for 7 s. Next, metallographic specimens from the longitudinal and cross sections were extracted along the length of the DS castings (Figure 1b) (from 5 to 80 mm from the metal/mold interface), grounded with silicon carbide papers from 100 to 1200 mesh and polished using diamond paste up to 1 μm. No chemical etching was needed to reveal the microstructure. Optical micrographs were taken through a GX41 Olympus Inverted Metallurgical Microscope (Olympus, Tokyo, Japan). The ImageJ software (1.8.0, National Institute of Health, Bethesda, MD, USA) was used to measure the dendritic arm spacings ($\lambda_1$, $\lambda_2$ and $\lambda_3$). At least 40 independent readings were taken for each selected position, with the average taken to be the local spacing. The triangle method was employed on transverse sections to quantify $\lambda_1$, while the linear intercept method was applied on longitudinal sections to measure $\lambda_2$. The measurements of $\lambda_3$ were performed on transverse sections using the intercept method. The area fraction (AF) of the second phase particles was also determined. The circularity (C)—a dimensionless shape factor of the IMC particles—was defined by a relation between the area of the particle surface (S) and its perimeter, i.e., $C = 4\pi S/p^2$ (according to the ASTM E562-11 [30] and ASTM E1382-97 [31] standards).

X-ray fluorescence (XRF) analysis was performed on a surface of 100 $mm^2$ for compositional analysis using a Rigaku device (model RIX3100) (Rigaku, Tokyo, Japan). X-ray diffraction patterns were obtained using a Panalytical X'Pert PRO MRD XL (Malvern Panalytical, Cambridge, UK) with a 2-theta range from 35 to 90 deg, Cu-K$\alpha$ radiation and a Ni filter with a wavelength of 0.15406 Å. XRD analyses were performed on three cross-sectioned samples with a voltage of 45 kV, a current of 40 mA and a scan interval of $20° \leq 2\theta \leq 90°$, in which the samples were rotating during the analyses. The IMCs were characterized using SEM micrographs taken via a ZEISS Scanning Electron Microscope (ZEISS-EVO-MA15) equipped with an Oxford X-Max model Energy Dispersive Spectrometer (EDS) (Zeiss Microscopy, Oberkochen, Germany).

### 2.3. Mechanical Tests

Vickers hardness tests were carried out using a Shimadzu HMV-2 model hardness measuring (Shimadzu, Kyoto, Japan) test device employing a 0.5 kgf load and a dwell time of 15 s, according to the ASTM E384 standard [32]. The hardness value adopted for each representative sample was the average of at least 20 measurements. Finally, the specimens were subjected to tensile tests using an MTS 810-FlexTest 40 machine (MTS Company, AL, USA), according to specifications of the ASTM Standard E 8M/04 [33] at a strain rate of $10^{-3}$ mm s$^{-1}$. For reproducibility purposes, the tensile tests were performed in triplicate. Samples for the tensile testing were taken every 5 mm from the cooled base (metal/mold interface) of the castings, i.e., 14 specimens were sectioned for each alloy casting (Figure 1b).

### 2.4. Multiple Linear Regression (MLR) Analysis

MLR analysis was used to determine the fits relating the mechanical properties (dependent variables) to the dendrite arm spacings (predictor variables). Such analysis was composed of three steps: (i) organization of the experimental dataset; (ii) setup and running of the MLR calculations and (iii) interpretation of the MLR analysis output according to the following criteria: significance F, *p*-values and $R^2$. Only variables with a significance F < 0.05 were considered. A *p*-value was considered statistically significant only if it was lower than 0.05. Finally, the fitness of the regression model was interpreted by the $R^2$ value

($R^2$ close to 1 indicates that a regression line fits the data well, while an $R^2$ near 0 indicates that the regression line does not fit the data very well).

## 3. Results and Discussion

### 3.1. Cooling Rate, Growth Rate and Macrosegregation

Figure 2a,b show the thermal profiles recorded during the directional solidification experiments carried out with the Al-7Si-(0.5 and 1)Mg alloys. Each curve corresponds to the thermal history measured at a specific position in the casting (P), considering that the P-axis starts at the heat-extracting surface—the point P = 0—and increases towards the interior of the DS casting. Note that there is a sharp drop in temperature for the thermocouples closer to the metal/mold interface, but this variation in temperature with time gradually decreases along the P-axis, i.e., along the length of the DS casting. Such behavior can be explained by analyzing the $V_L$ and $\dot{T}_L$ plots shown in Figure 2c,d, respectively. It can be seen that higher values of these thermal parameters are associated with positions nearer to the cooled bottom of the casting. This can be attributed to the good surface conformity between the solidified shell and the mold surface during the initial stages of solidification. However, the decrease in $V_L$ and $\dot{T}_L$ as the solid grows can be attributed to two main reasons: (1) the rise of the thermal resistance corresponding to the solidified shell and (2) the air gap formation at the casting/mold interface, since it hinders the interfacial heat transfer [34]. For a given position from the metal/mold interface, the Al-7Si-1Mg alloy is characterized by higher $V_L$ and $\dot{T}_L$ values compared with the corresponding values of the other investigated alloy. Thus, it seems like the increase in Mg content facilitates the efficiency of heat extraction during solidification either though a more efficient thermal contact at the metal/mold interface or though the appropriate modification of the alloy's thermophysical properties. The same happens with the Al-(1.2 and 7) Mg-1.5Fe alloys, as reported by Gomes et al. [35].

The liquid/solid transformation is accompanied by the release of thermal energy, with a moving boundary separating the two phases of distinct thermophysical properties. The analysis of heat transfer during solidification has essentially two objectives: the determination of the temperature distribution and that of the growth/cooling rates along the length of the castings. During the cooling of the castings, the main modes of heat extraction are: (i) thermal conduction in the metal and in the mold; (ii) Newtonian heat transfer at the metal/mold interface; (iii) forced convection of the cooling liquid at the mold bottom. Considering the position 85 mm from the metal/mold interface for both cooling curves presented in Figure 2a,b, the times of passage of the solidus isotherm by such position in the castings were 285 s and 220 s for the Al-7Si-0.5Mg and Al-7Si-1Mg alloy castings, respectively.

Table 2 shows the Si, Mg and Fe solute concentrations along the length of the DS castings. As can be seen, the Mg content remained almost the same and close to the nominal compositions. On the other hand, Si presents a minimal normal macrosegregation. Note that the Si concentration tends to become higher towards the interior of the castings. This means that Si is rejected into the melt as the solid/liquid interface advances, thus resulting in higher Si concentrations for positions further away from the cooling base. One possible reason for this is the fact that solidification becomes slower along the P-axis, thus providing more time for the enrichment of the local Si concentration. Even though such behavior is different from that previously observed in DS Si-containing Al alloy castings [16,36,37], it is quite similar to that found for a ternary Al-15Si-1.5Mg alloy processed under a melt superheating of 21% [38]. Kakitani et al. [38] also attributed the Si normal macrosegregation to slower solidification conditions, which were represented by the eutectic growth rate ($\lambda_{EUT}$). It is worth mentioning that the $\lambda_{EUT}$ values reported by Kakitani et al. [38] lie between 0.2 and 1.4 mm/s. It can also be seen in Table 2 that the Fe concentration along the P-axis presents a constant behavior around ~0.1wt.%. Such residual traces of Fe are able enough to form Fe-rich phases such as $\beta$-AlFeSi IMCs.

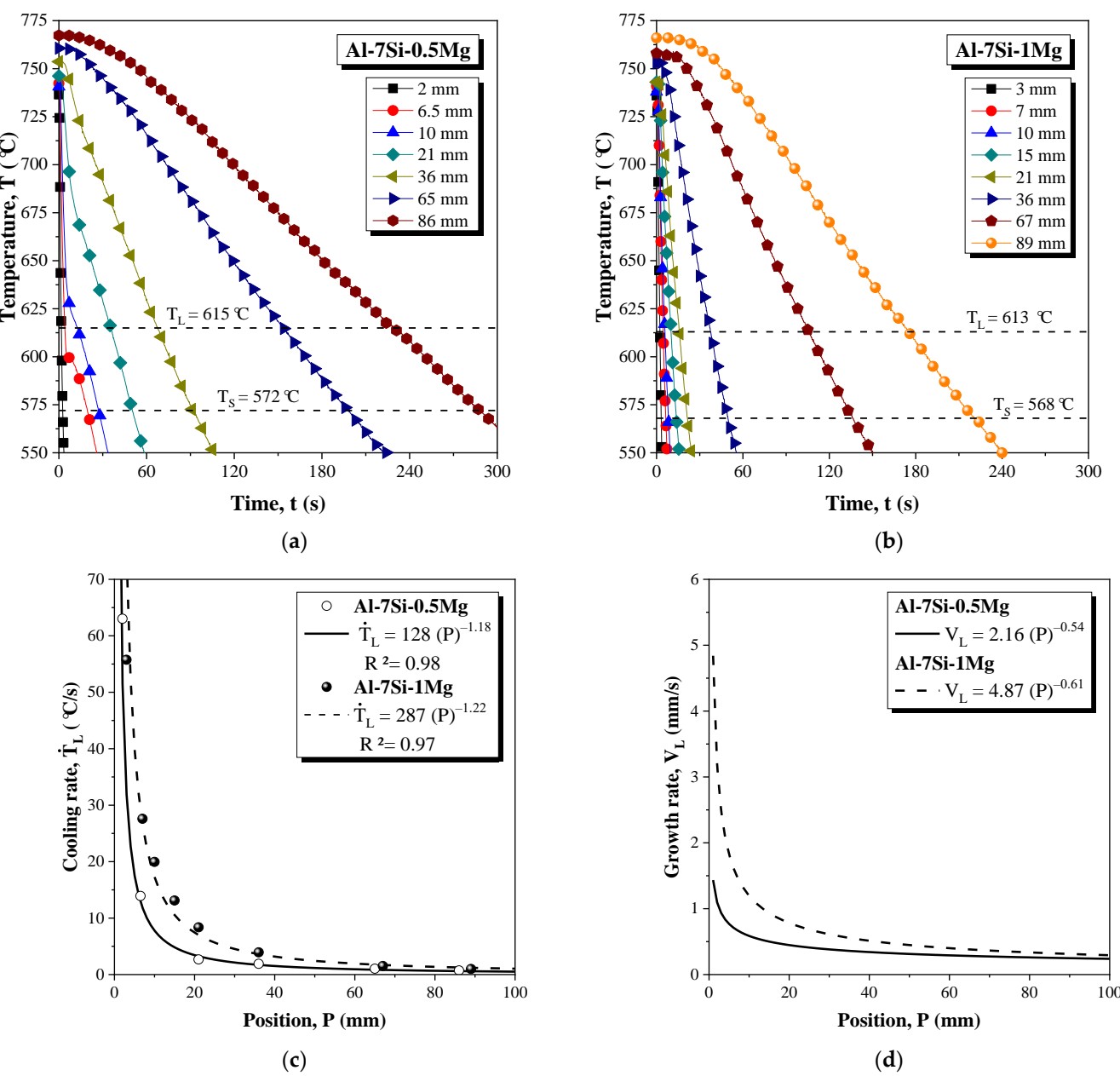

**Figure 2.** Cooling curves of the (**a**) Al-7Si-0.5Mg and (**b**) Al-7Si-1Mg alloys, and profiles of the (**c**) cooling rate and (**d**) growth rate along the length of the DS castings.

**Table 2.** Si, Mg and Fe solute concentrations along the length of the directionally solidified castings.

| Element | Alloy (wt.%) | 5 mm | 25 mm | 50 mm | 70 mm | 90 mm |
|---|---|---|---|---|---|---|
| Si | Al-7Si-0.5Mg | 6.67 | 6.30 | 7.48 | 7.41 | 7.67 |
| | Al-7Si-1Mg | 6.48 | 6.56 | 7.18 | 7.55 | 8.33 |
| Mg | Al-7Si-0.5Mg | 0.58 | 0.55 | 0.57 | 0.54 | 0.55 |
| | Al-7Si-1Mg | 1.09 | 0.98 | 0.95 | 0.97 | 0.96 |
| Fe | Al-7Si-0.5Mg | 0.15 | 0.14 | 0.15 | 0.14 | 0.14 |
| | Al-7Si-1Mg | 0.13 | 0.11 | 0.12 | 0.11 | 0.10 |

A process of separation diffusion occurs at the solid/liquid interface of the growing dendrites, giving rise to a chemical inhomogeneity from the center to the surface of the dendrite arms. Such a phenomenon has initially been called dendrite liquation, although the

term microsegregation is more commonly used in the literature [39]. The local solidification cooling rate along the length of the casting affects the composition in the solidification range, i.e., the increase in the cooling rate leads to a progressive increase in segregation and thus in the amount of non-equilibrium eutectics [40]. Although the phenomenon of microsegregation is outside the scope of the present work, it is worth noting that the solidification thermal parameters, i.e., the cooling and growth rates of the present study, may have an important role in the quantification of microsegregation. As shown by Meza et al. [41] in a study on the directional solidification of the Al-Fe and Al-Cu alloys, the experimental microsegregation profiles from the central parts of the dendrites (Al-Cu) and cellular (Al-Fe) cores were shown to vary with the growth rate from the central part of the dendrite/cell cores to the limit of the intercellular/interdendritic regions.

### 3.2. Macrostructure, Microstructure and Microstructural Spacing

Figure 3 presents the macrostructures and representative optical microstructures of the transverse and longitudinal sections at three different cooling rates for both studied alloys. Note that columnar grains prevailed along the length of the DS castings with no trace of equiaxed grains, thereby confirming the directionality of the solidification process without any macrostructural transition. Regarding the optical micrographs, Al-rich dendrites with a Maltese-cross morphology can clearly be seen. Consequently, it is possible to measure dendritic arm spacings along the casting and to correlate them with both $V_L$ and $\dot{T}_L$. Besides that, it is noted that a finer microstructure for the positions closer to the cooling base results in higher $V_L$ and $\dot{T}_L$ values. However, the microstructures become coarser as the values of these higher thermal parameters decrease.

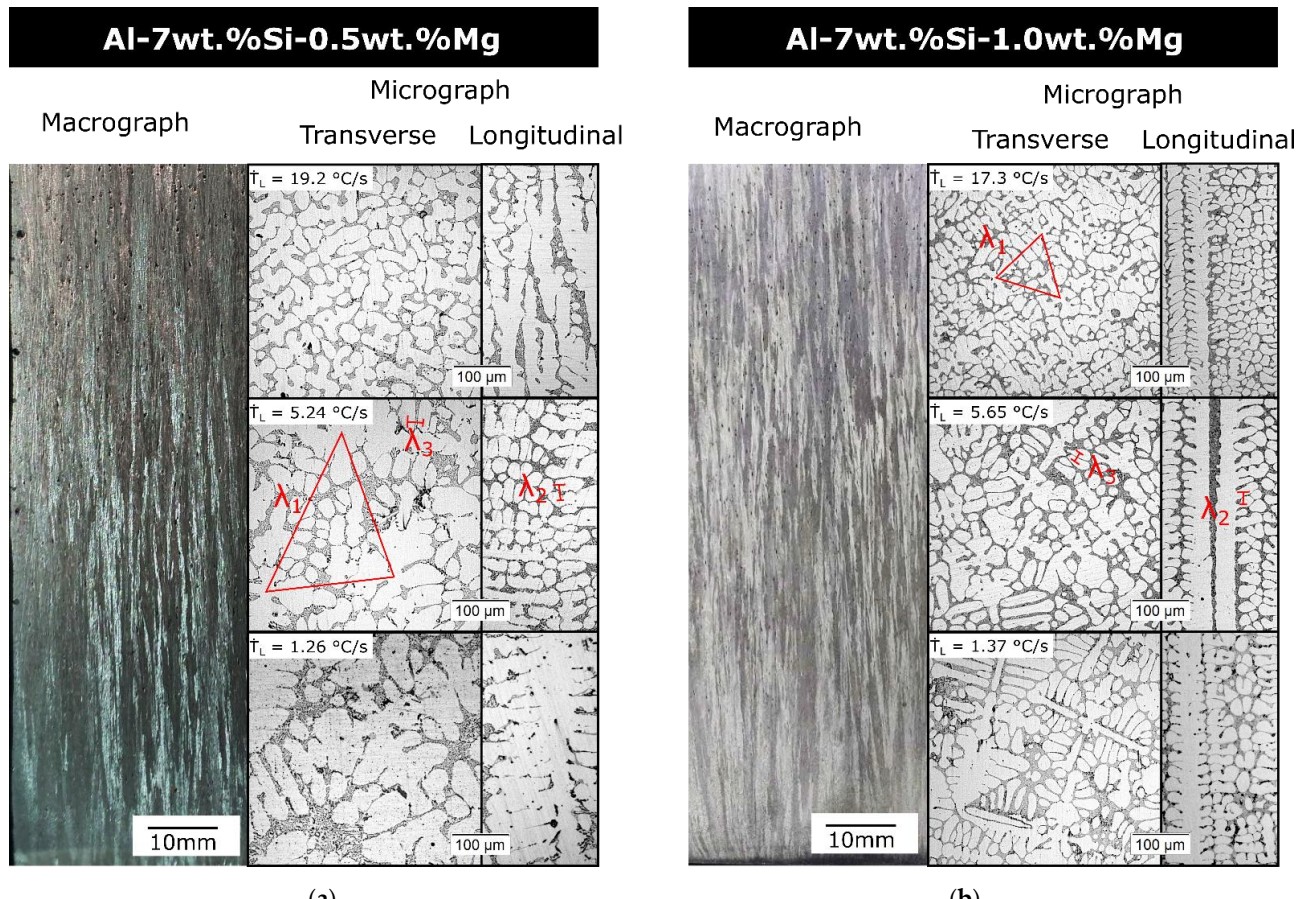

**Figure 3.** Solidification macrostructure and characteristic microstructures of the (**a**) Al-7Si-0.5Mg and (**b**) Al-7Si-1Mg alloys for three different $\dot{T}_L$ values. The red lines represent each measured dendritic spacing carried out: primary ($\lambda_1$), secondary ($\lambda_2$) and tertiary ($\lambda_3$).

As can be seen in Figure 4a–d, the evolution of $\lambda_1$, $\lambda_2$ and $\lambda_3$ in terms of $V_L$ and $\dot{T}_L$ follows the same power-type functions for both studied alloys, with $R^2$ values > 0.9. In a preliminary view, the addition of 0.5Mg to the Al-7Si-0.5Mg alloy seems to not affect the coarsening of the dendritic arms. However, in order to conduct a proper comparative analysis with the literature, experimental equations proposed in previous works carried out with Al-Si-based alloys were also plotted in Figure 3. Note that the exponents were set to be the same values previously reported for several Al-based alloys [42–44]. Regarding $\lambda_1$, Silva et al. [27] proposed the equation $\lambda_1 = 263\,\dot{T}_L^{-0.55}$ for an Al-7Si-3Mg alloy. On the other hand, Peres et al. [44] observed that the expression $\lambda_1 = 220\,\dot{T}_L^{-0.55}$ better represented their experimental data for the Al-7Si alloy. When these functions are compared with the one proposed herein ($\lambda_1 = 328\,\dot{T}_L^{-0.55}$), it can be noticed that the addition of Mg to the Al-7Si alloy tends to promote higher $\lambda_1$ values. Such tendency is intensified when the Mg solute content is increased to 0.5, so much so that the multiplier of the equation becomes 328. However, it stabilizes when the Mg solute content achieves 1. Thus, increasing the Mg solute content in Al-7Si-Mg alloys will not necessarily always increase the $\lambda_1$ values. As can be seen, for a given $\dot{T}_L$ value, the Al-7Si-3 Mg presents a $\lambda_1$ lower than that of the Al-7Si-1 Mg alloy. This indicates that the tendency to increase $\lambda_1$ is mitigated at some Mg solute contents and may start behaving in the opposite way. Although $\lambda_1$ is mostly evaluated as a function of $\dot{T}_L$, determining its correlation with $V_L$ is just as important. That is why an experimental equation relating $\lambda_1$ to $V_L$ is proposed.

Regarding the behavior of $\lambda_2$ in Figure 4, it is worth pointing out that the –2/3 exponent is the same as that proposed by the Bouchard–Kirkaldy model [45]. This exponent was also shown to effectively fit the $\lambda_1$ dataset obtained by Peres et al. [44] and Silva et al. [27] for the Al-7Si and Al-7Si-3Mg alloys, respectively. The resulting equations were included in Figure 4d. While the $\lambda_1$ values of the Al-7Si alloy tended to increase with the addition of Mg, the opposite happened with $\lambda_2$. In fact, $\lambda_2$ decreases by about 42% when Mg is added to the Al-7Si alloy. However, no expressive difference is observed between the $\lambda_2$ of the Al-7Si-(0.5 and 1) Mg alloys. Furthermore, the equation proposed by Silva et al. [27] for the growth of $\lambda_2$ in the Al-7Si-3Mg alloy fits quite well the $\lambda_2$ values found in this work. Besides evaluating $\lambda_2$ as a function $V_L$, it is just as important to establish its correlation with $\dot{T}_L$. Hence, a $\lambda_2 = f(\dot{T}_L)$ equation is proposed based on our experimental measurements.

The tertiary dendritic ramifications present a fascinating behavior: they start growing when $V_L$ and $\dot{T}$ are lower than 0.62 mm/s and 6 °C/s, respectively. The $\lambda_3 = f(\dot{T})$ equation found herein is characterized by the –0.25 exponent, which is 2.2x less than −0.55, the exponent typically used in previous works [42–45]. The exponent -0.25 is the same as that used for the correlation of eutectic spacings. Silva et al. [46] observed that the use of the Jackson/Hunt law [47] for eutectic spacing can be extended to represent the growth of primary and tertiary arm spacing due to a substantial formation of the eutectic constituent. Although the equation proposed by Silva et al. [27] for the growth of $\lambda_3$ in the Al-7Si-3Mg alloy is represented by the exponent –0.55, it provides values closer to the experimental dataset found for the studied alloys.

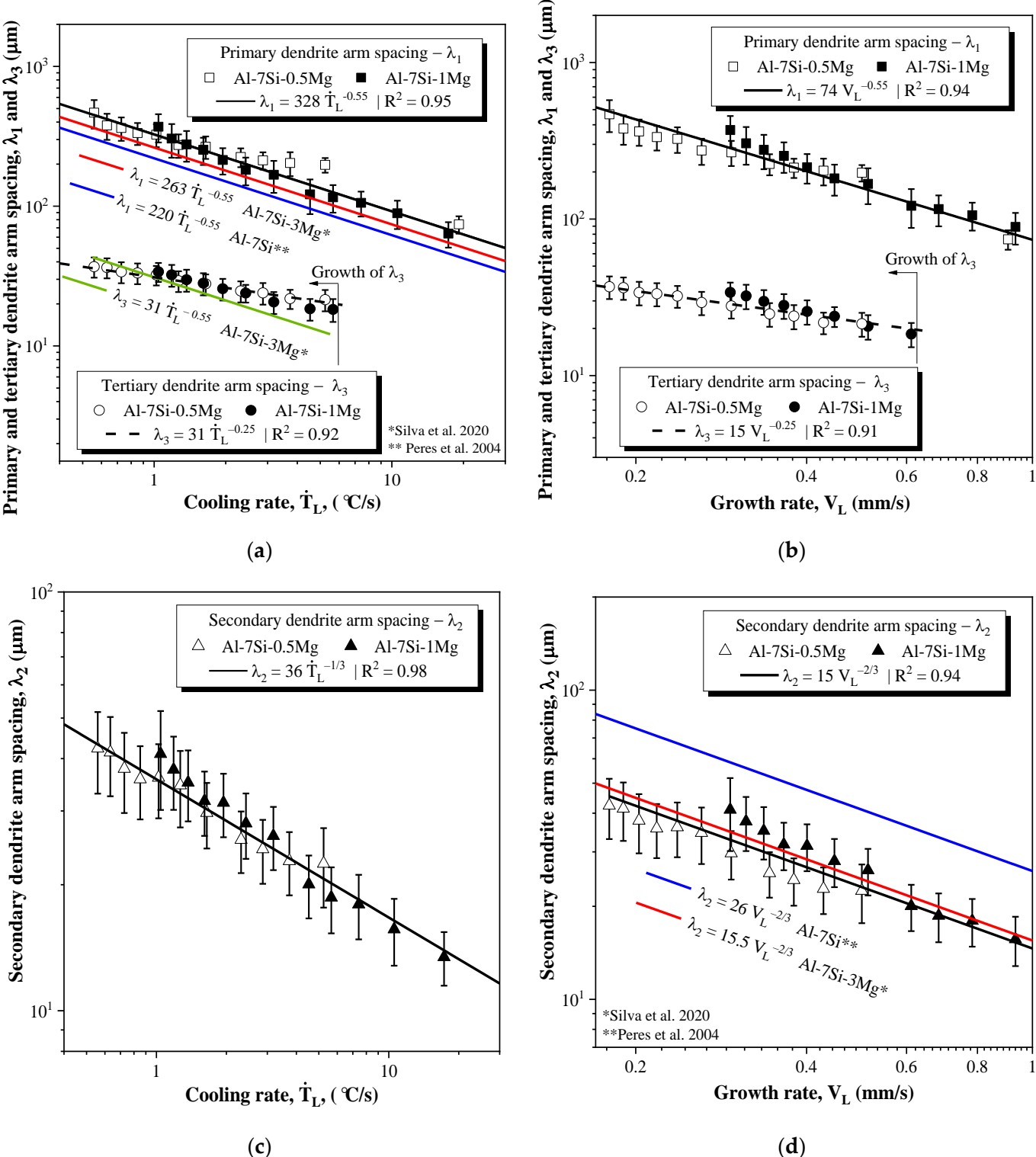

**Figure 4.** Correlations of the dendrite arm spacings with $V_L$ and $\dot{T}_L$ for the ternary Al-7wt.%Si-xMg alloys casting. (**a**) $\lambda_1$ and $\lambda_3$ evolution as a function of cooling rate; (**b**) $\lambda_1$ and $\lambda_3$ evolution as a function of growth rate; (**c**) $\lambda_2$ evolution as a function of cooling rate; and (**d**) $\lambda_2$ evolution as a function of growth.

### 3.3. Intermetallic Compound (IMC) Analysis

Figure 5 shows the Scheil–Gulliver solidification paths generated by the Thermo-Calc (TTAL5 database) software for the studied Al-Si-Mg alloys, considering Fe at a residual

trace of 0.15wt.%. It can be noted that the phases form according to the same sequence. In both cases, four phases are expected to compose the final microstructure as follows: α-Al, Si, π-AlSiFeMg and Mg$_2$Si. As a result of its higher Mg solute content, the Al-7Si-1Mg alloy is expected to possess a higher Mg$_2$Si fraction due to the existence of more Mg to react with Si. It is worth mentioning that Fe-rich IMCs can compromise the mechanical properties of these alloys, especially acicular β-AlSiFe IMCs. Some studies confirm this deleterious effect, such as the work of Kashyap et al. [23]. They found out that the reduction in the mechanical strength of an Al-7Si-Mg alloy can be directly linked to the increase in Fe content (from 0.02 to 0.8). From the Scheil–Gulliver simulations carried out, it can be inferred that, for the Mg contents studied, the entire β-AlSiFe phase will be converted into π-AlSiFeMg, so the observed transformation from β-AlSiFe into π-AlSiFeMg can be beneficial due to the change in the morphology of the Fe-rich intermetallics for a less harmful shape [48,49].

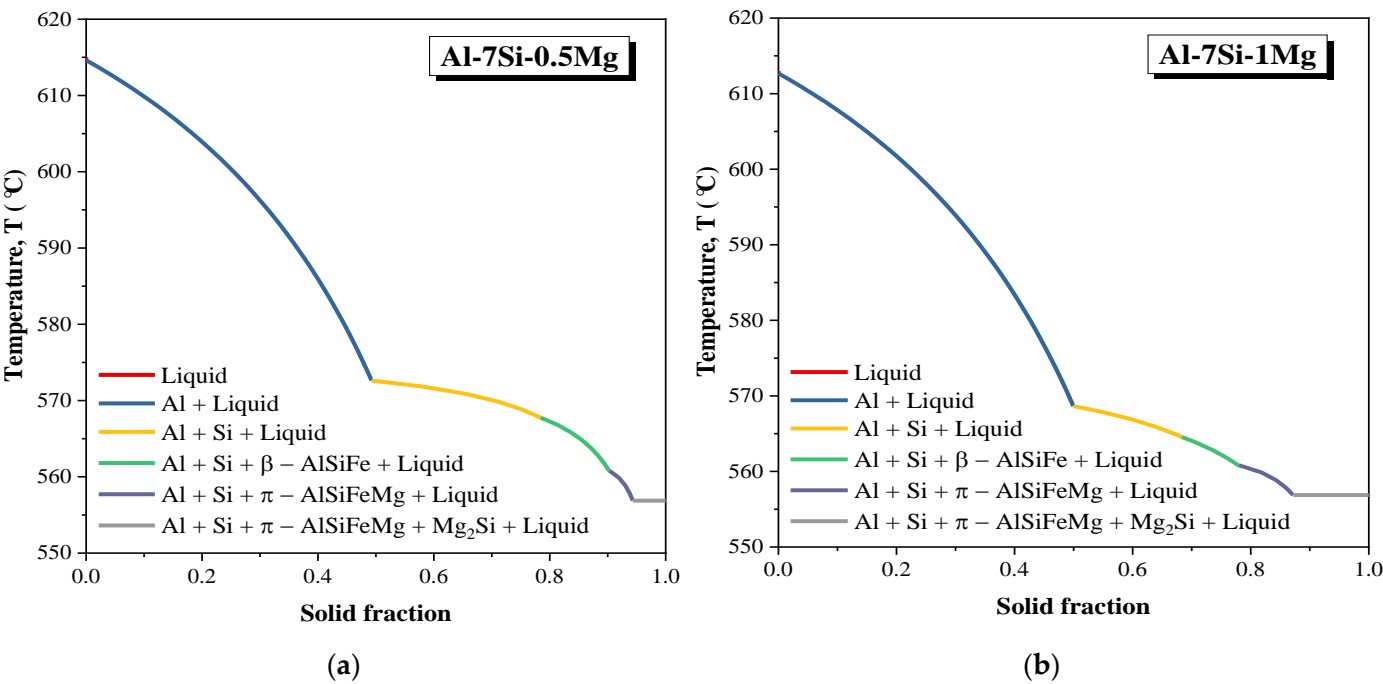

**(a)** **(b)**

**Figure 5.** Scheil solidification path of the (**a**) Al-7Si-0.5Mg and (**b**) Al-7Si-1Mg alloys.

The XRD patterns shown in Figure 6 that are associated with the SEM/EDS analysis (Figure 7) confirm the presence of the four phases predicted in the solidification paths, i.e., α-Al, Si, π-AlSiFeMg and Mg$_2$Si. The primary α-Al phase is shown to constitute the dendrites, while the eutectic constituent is composed of a complex eutectic mixture of the α-Al, Si, Mg$_2$Si and π-AlSiFeMg phases. It can be seen that the Mg$_2$Si particles present in the interdendritic regions essentially possess an acicular morphology, as well as some Fe-rich IMCs, which is in accordance with previous studies [22,50]. As a matter of fact, such microstructure is basically the same as that reported by Lima et al. [5] for a horizontally solidified Al-7Si-0.3Mg alloy. Interestingly, in this work, it can be noticed that the Si particles become finer with the increase in the alloy's Mg content from 0.5 to 1. As confirmed in previous studies, Al-7Si-Mg alloys solidified at lower cooling rates may present Si particles with an acicular morphology [6,22,50].

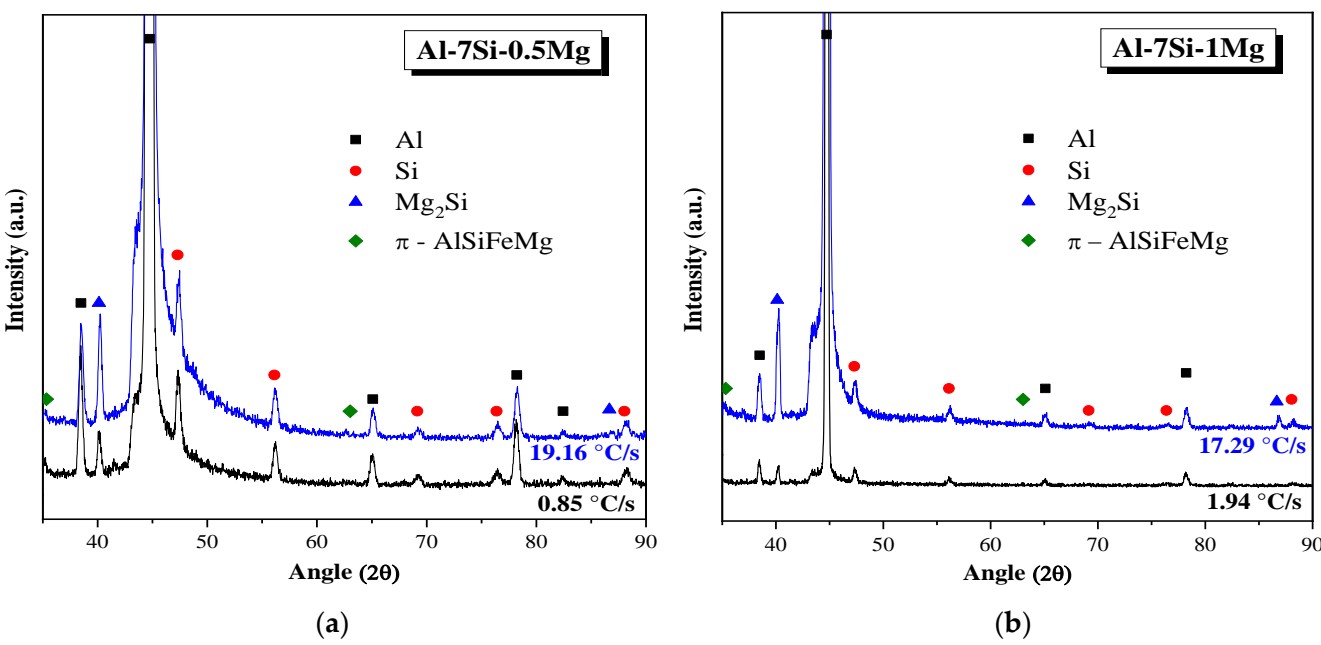

**Figure 6.** XRD patterns for the (**a**) Al-7Si-0.5Mg and (**b**) Al-7Si-1Mg alloys.

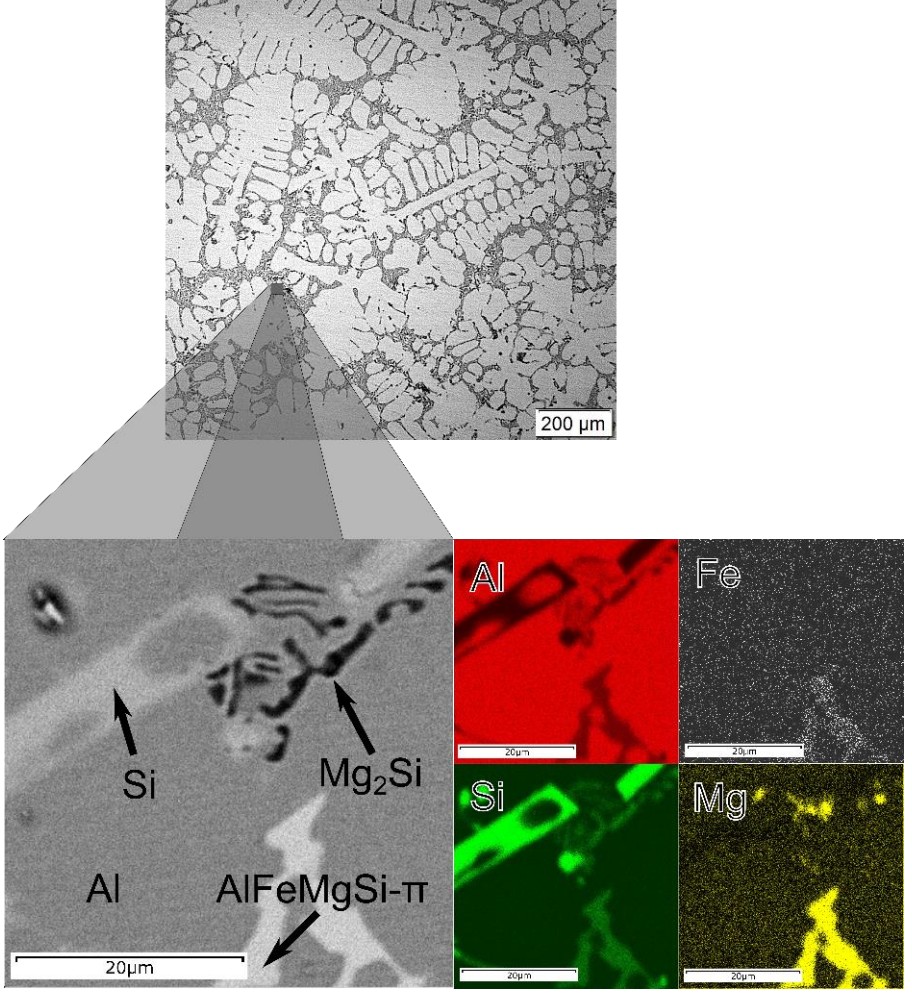

**Figure 7.** Characteristic SEM micrograph with an EDS analysis showing the microstructure phases.

It can also be seen in Figure 8 that the area fraction of the AlSiFeMg IMCs in the Al-7Si-1Mg alloy is lower than that pertaining to the Al-7Si-0.5Mg alloy for medium and faster cooling. Interestingly, this changes for slow cooling, i.e., such area fraction becomes comparatively higher for the Al-7Si-1Mg alloy. According to Silva et al. [27], the Al-rich matrix of Al-Si-Mg alloys solidified at higher cooling rates contains more Mg as compared with that resulting from slower cooling. Thus, it is inferred that the Al-7Si-1Mg solidified at faster cooling conditions presents less deleterious AlFeSiMg IMCs. Assuming that the area fraction and volume fraction are equal, the mass fraction of each phase can be estimated using the following densities (g/cm$^3$): 2.70 for Al [51], 2.32 for Si [52], 1.99 for Mg$_2$Si [53] and 2.83 for $\pi$-AlFeSiMg [54]. Furthermore, the theoretical mass fraction of each phase can be determined from the plots of the equilibrium phase fractions shown in Figure 9, considering the values at room temperature. A comparative analysis for these mass fractions is given in Table 3, where a relatively good agreement between the experimental and theoretical values pertaining the $\pi$-AlFeSiMg phase can be seen. Regarding the mass fraction of the Mg$_2$Si phase, the theoretical values overestimate the experimental data, especially for the Al-7Si-1Mg alloy. A possible reason for this is that the cooling conditions studied herein, even the slower one, lie far from the equilibrium. Thus, there is no time to complete the diffusive processes. It is worth mentioning that the faster the solidification, the better the suppression of the Mg diffusion [38]. Therefore, Mg can be retained in the Al-rich matrix, thus hindering the formation of Mg$_2$Si IMCs in the interdendritic regions.

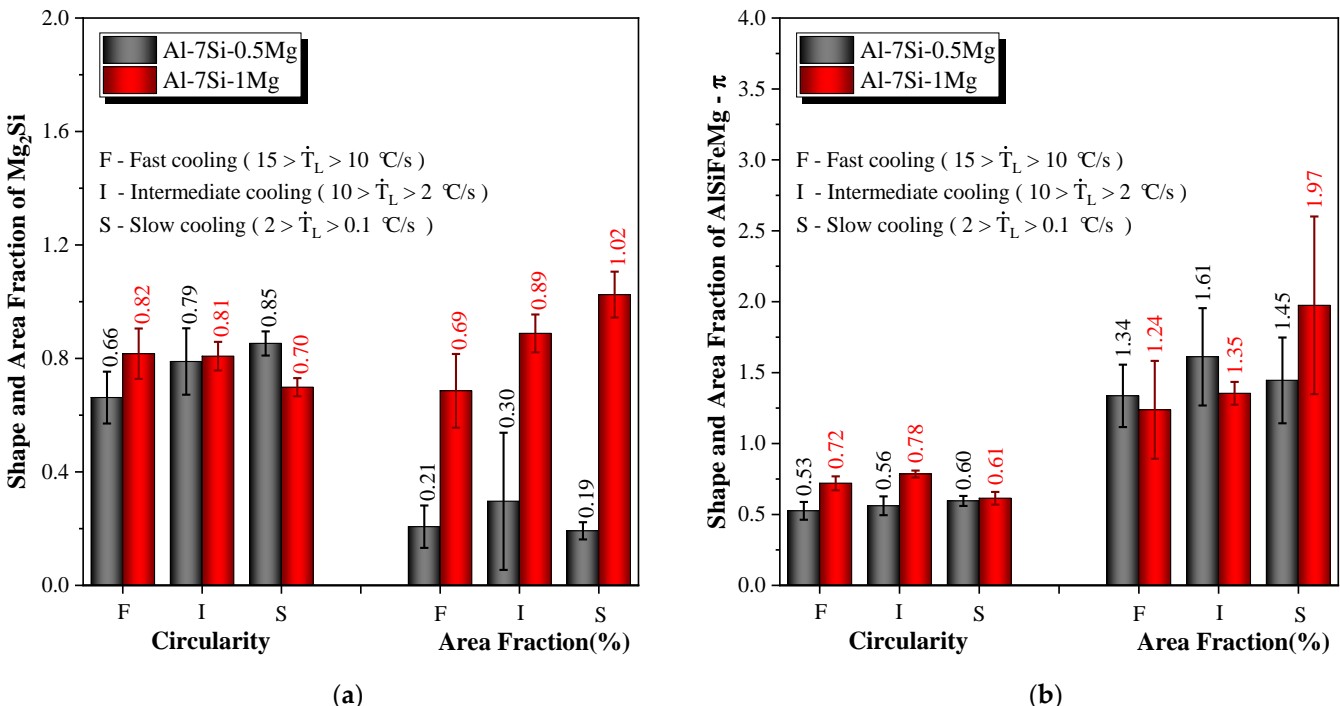

**Figure 8.** Column graphs of the circularity and area fraction of the (**a**) Mg$_2$Si and (**b**) Al-Si-Fe-Mg IMCs in the Al-7Si-(0.5 and 1) alloys for three different ranges of cooling rates.

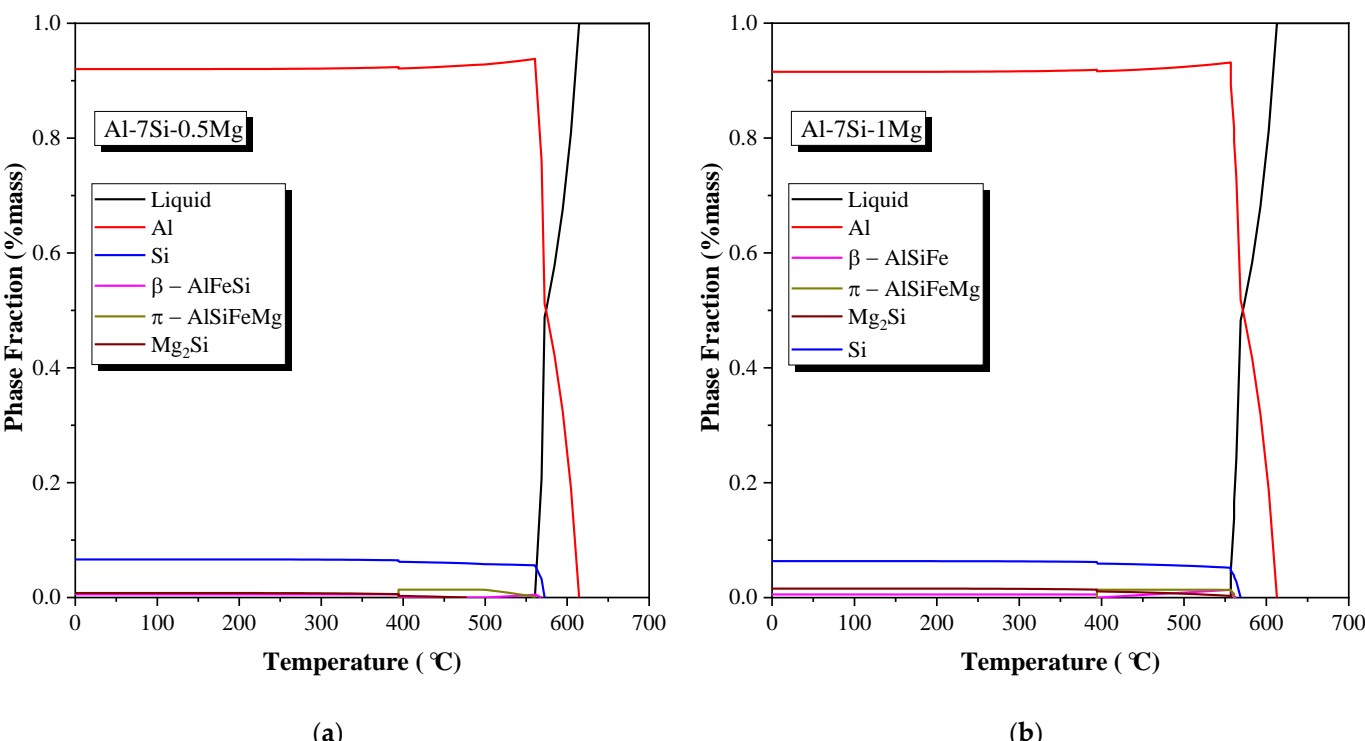

**(a)** **(b)**

**Figure 9.** Equilibrium phase fraction plots generated by the Thermo-Calc (with the TTAL5 database) software for the (**a**) Al-7Si-0.5Mg and (**b**) Al-7Si-1Mg alloys.

**Table 3.** Experimental and theoretical values of the mass fractions of the microstructure phases for the studied alloys.

| Alloy | Phase | Experimental | | Calculated from Thermo-Calc |
|---|---|---|---|---|
| | | Cooling Regime | Mass Fraction (wt.%) | Equilibrium Mass Fraction (wt.%) |
| Al-7Si-0.5Mg | $Mg_2Si$ | Fast | 0.002 | 0.01 |
| | | Intermediate | 0.002 | |
| | | Slow | 0.002 | |
| | $\pi$-AlFeSiMg | Fast | 0.015 | 0.01 |
| | | Intermediate | 0.018 | |
| | | Slow | 0.016 | |
| Al-7Si-1Mg | $Mg_2Si$ | Fast | 0.005 | 0.02 |
| | | Intermediate | 0.007 | |
| | | Slow | 0.008 | |
| | $\pi$-AlFeSiMg | Fast | 0.014 | 0.01 |
| | | Intermediate | 0.015 | |
| | | Slow | 0.022 | |

Figure 10 presents the ultimate tensile strength, elongation and hardness of the analyzed alloys considering the same three cooling regimes. The high values of such mechanical properties at the positions associated with high cooling rate values are attributed to the refinement of the microstructure. The only exception is the hardness of the Al-7Si-0.5Mg alloy, which seems to be unaffected. Overall, it can be observed that the Al-7Si-1Mg alloy presents a better mechanical response compared with that of the Al-7Si-0.5Mg alloy. This is because the Al-7Si-1Mg alloy presents a quantity of $Mg_2Si$ that is 3 to 5 times higher than that of the Al-7Si-0.5Mg alloy for all cooling conditions. According to Zoloterevsky

et al. [12], the hardness of the Mg$_2$Si phase is 5.36 GPa (546.5 HV) at room temperature. This means that such phase is very effective in the strengthening of Al-Si-Mg alloys.

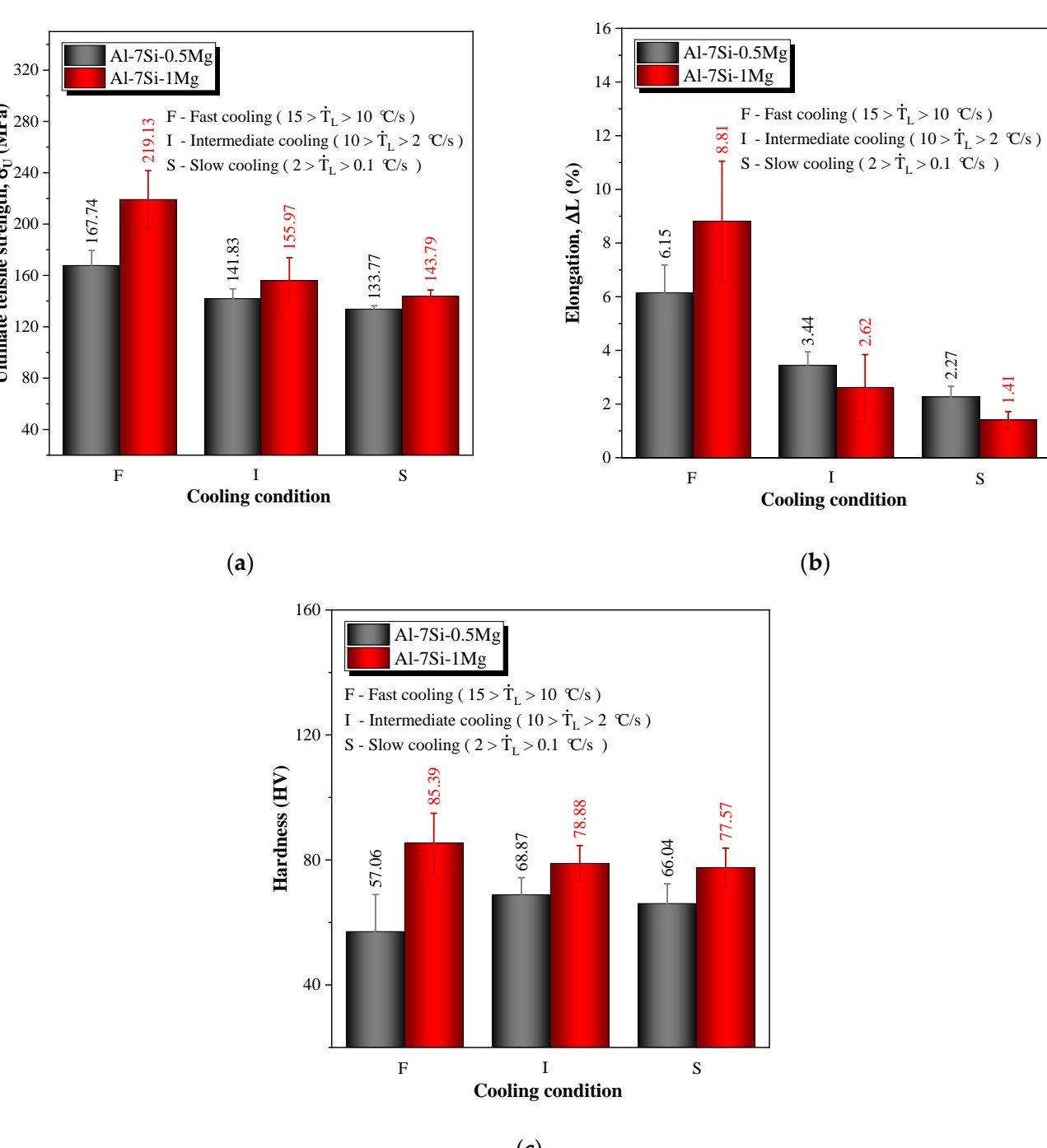

**Figure 10.** (**a**) Ultimate tensile strength, (**b**) elongation and (**c**) hardness of the Al-7Si-0.5Mg and Al-7Si-1Mg alloys considering three different cooling regimes.

### 3.4. Mechanical Behavior and MLR Analysis

MLR plots relating the tensile properties and hardness of the investigated Al-7Si-(0.5 and 1) Mg alloys to the inverse of the square root of $\lambda_1$, $\lambda_2$ and $\lambda_3$ (in terms of Hall–Petch-type equations) are given in Figure 11. Additional horizontal axes were included in the MLR graphs in order to have all the dendritic spacings represented. As a consequence, the linear behavior of each profile was changed into a curve one. This is because some points seem to not match perfectly with the curve, so the determination coefficients ($R^2$)

are > 0.77. It is worth mentioning that changes in any dendritic spacing are directly associated with shifts in the others, which might promote multicollinearity during the MLR analysis. However, multicollinearity does not interfere in the usefulness of the regression equations or the reliability of the model herein used, as all MLR equations present $R^2$ values > 0.77. As previously explained, the hardness of the Al-7Si-0.5Mg alloys was around 67 HV, with no significant changes. That is why no MLR fit was proposed for this particular case. Compared with the SLR equations, the MLR ones are characterized by higher $R^2$ values. In addition, MLR equations have the advantage that all the data are retained, and the predicted endpoint may be better estimated. However, eventually, overfitting may occur with the addition of independent variables (the number of microstructural spacing types), decreasing the accuracy of the model. It can be observed that the increase in the Mg alloy content from 0.5 to 1 was shown to promote an increase for both the ultimate tensile strength ($\sigma_u$) and elongation. The $\sigma_u$ values of the Al-7Si-1Mg alloy are higher than those of the Al-7Si-0.5Mg alloy. On the other hand, the elongation of the alloy with 1Mg is higher than that of the Al-7Si-0.5Mg alloy only for $\lambda_2^{-1/2}$ values higher than 0.22 $\mu m^{-1/2}$. Li et al. [22] showed that: (1) the increase in the Mg solute content increases the ultimate tensile strength in a composition range between 0 and 3wt.%, and (2) the elongation increases with the increase in the alloy Mg content in the range from 0 to 1wt.% due to the formation of $Mg_2Si$ IMCs. Therefore, the increase in Mg in alloys of the Al-7Si-XMg system until 1 is shown to improve their mechanical properties such as hardness, $\sigma_u$ and elongation because of: (1) the increase in the fraction of $Mg_2Si$ IMCs; (2) the suppression of the formation of Fe-rich IMCs and (3) morphological changes in the microstructural particles.

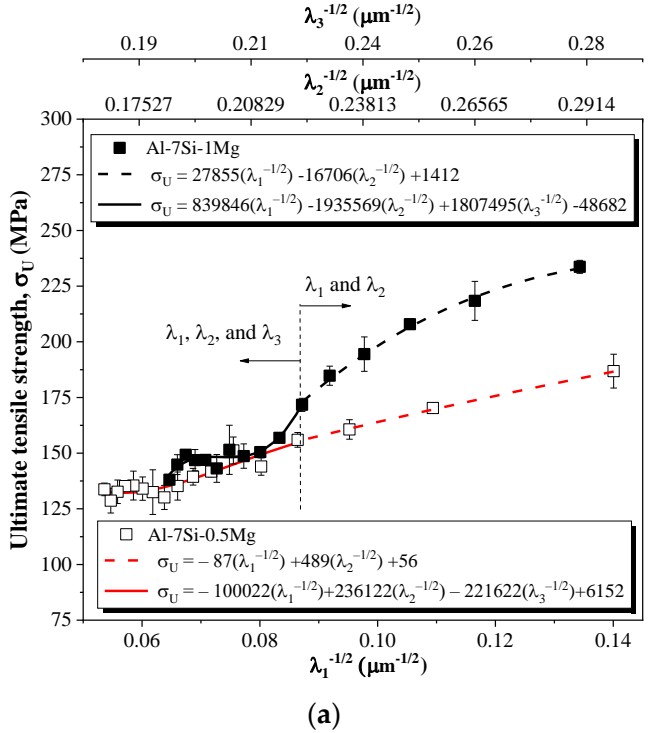

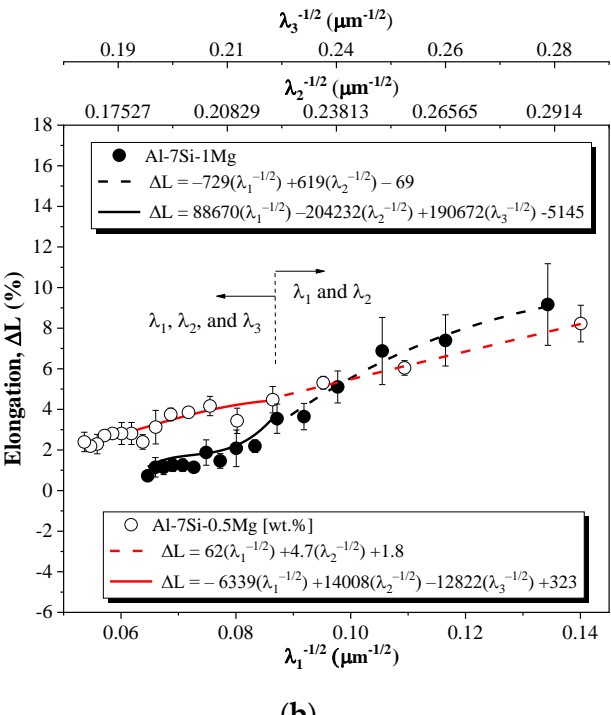

(a)                               (b)

**Figure 11.** *Cont.*

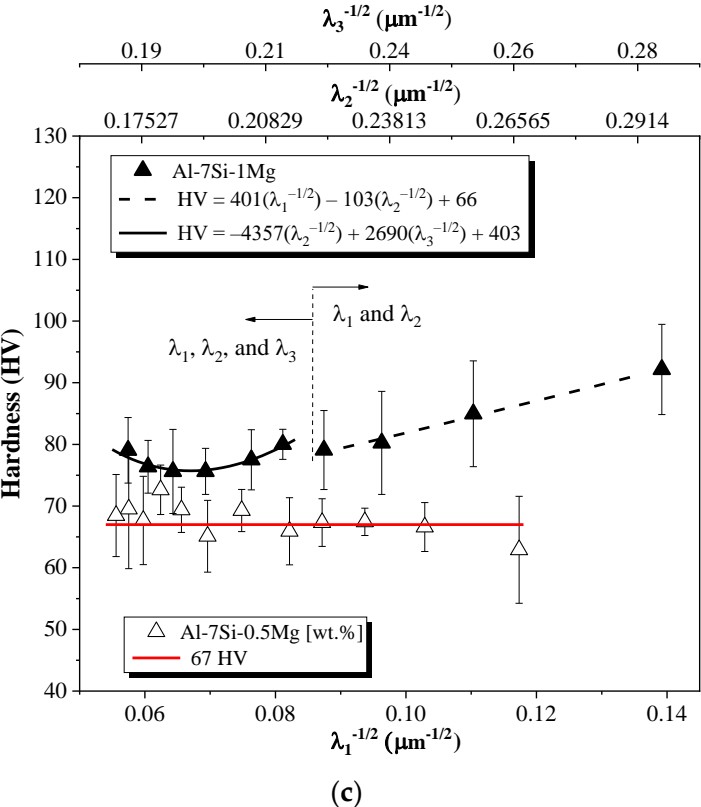

**(c)**

**Figure 11.** MLR plots for both Al-7Si-0.5Mg and Al-7Si-1Mg alloys: (**a**) ultimate tensile strength (**b**) elongation and (**c**) hardness.

## 4. Conclusions

- The macrostructures of both the Al-7Si-0.5Mg and Al-7Si-1.0Mg DS castings were shown to be characterized by columnar grains with no trace of equiaxed grains, thereby confirming the directionality of the solidification process. The microstructures were shown to be formed by Al-rich dendrites with a Maltese-cross morphology, with a complex eutectic mixture of Si particles and $\pi$-AlSiFeMg/$Mg_2Si$ intermetallics distributed in the interdendritic regions.

- Experimental power-type functions have been derived relating the interdendritic spacings $\lambda_1$, $\lambda_2$ and $\lambda_3$ to $\dot{T}_L$ and $V_L$:

$$\lambda_1 = 328\,\dot{T}_L^{-0.55} \text{ and } \lambda_1 = 74\,\dot{V}_L^{-0.55}$$

$$\lambda_2 = 36\,\dot{T}_L^{-1/3} \text{ and } \lambda_2 = 15\,\dot{V}_L^{-2/3}$$

$$\lambda_3 = 31\,\dot{T}_L^{-0.25} \text{ and } \lambda_3 = 15\,\dot{V}_L^{-0.25}$$

where $\lambda$ [µm], $\dot{T}_L$ [K/s] and $V_L$ [mm/s].

- These expressions were shown to be valid for both the Al-7Si-0.5Mg and Al-7Si-1.0Mg alloys, i.e., the increase in Mg content from 0.5wt% to 1.0wt.% has not affected the coarsening of the dendrite arms.

- The highest values of the ultimate tensile strength, the elongation to fracture and the hardness are shown to be associated with high solidification cooling rates and consequently related to the refinement of the microstructure. The only exception is the hardness of the Al-7Si-0.5Mg alloy, which seems to be unaffected. Overall, it was shown that the Al-7Si-1Mg alloy presents a better mechanical response as compared with that of the Al-7Si-0.5Mg alloy.

- MLR plots relating the tensile properties (ultimate tensile strength and elongation to fracture) and hardness of the Al-7Si-(0.5 and 1) Mg alloys to the inverse of the square root of λ1, λ2 and λ3 (in terms of Hall–Petch-type equations) were developed.

**Author Contributions:** Conceptualization, C.S. and C.B.; methodology, C.S.; A.B.; T.V.; and C.B.; software, C.S.; and A.B.; validation, C.S.; A.B.; T.V.; D.A.P.R. and C.B.; formal analysis, C.S.; A.B. and T.V.; investigation, C.S.; resources, C.S.; data curation, C.S.; writing—original draft preparation, C.S.; A.B.; N.C. and C.B.; writing—review and editing, A.G.; N.C. and C.B.; supervision, C.B.; project administration, C.S. and C.B.; funding acquisition, C.B. All authors have read and agreed to the published version of the manuscript.

**Funding:** National Council for Scientific and Technological Development: 407871/2018-7.

**Institutional Review Board Statement:** Not applicable.

**Informed Consent Statement:** Not applicable.

**Data Availability Statement:** Not applicable.

**Acknowledgments:** The authors acknowledge CNPq—the National Council for Scientific and Technological Development (Grant: 407871/2018-7)—and CAPES—Coordenacão de Aperfeicoamento de Pessoal de Nível Superior, Brazil—for the financial support, as well as the Brazilian Nanotechnology National Laboratory—LNNano—for the use of the X-ray diffractometer.

**Conflicts of Interest:** The authors declare no conflict of interest.

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
