# Peer review of "Assessing Microstructure Tensile Properties Relationships in Al-7Si-Mg Alloys via Multiple Regression"

_metals, doi:10.3390/met12061040_

Round 1
Reviewer 1 Report
This is a comprehensive experimental work on microstructure and strength change during solidification. Minor revisions are suggested as follows:
Ÿ It would be good to show an actual photo of experimental setup and sample.
Ÿ Fig. 2 (a), why is there only four data points for 6.5 mm curve? There is a change of slope at around 600oC.
Author Response
Comments and Suggestions for Authors
This is a comprehensive experimental work on microstructure and strength change during solidification. Minor revisions are suggested as follows:
It would be good to show an actual photo of experimental setup and sample.
Thank you very much for the comment. Figure 1 has been updated, in order to present the solidification apparatus, as well as the sectioning of the samples for metallographic and tensile tests.
Fig. 2 (a), why is there only four data points for 6.5 mm curve? There is a change of slope at around 600oC.
Thank you very much for the comment. All thermal cooling data were collected at an acquisition rate of 5 points per second. In Figure 2, the curves were edited with a view to reducing point pollution, so it may appear that they had different acquisition rates, but this did not happen. This information was added to Materials and Methods (page 3, lines 138 and 139).

Reviewer 2 Report
In general, the study was performed at a good level. The presented results of the performed experiments have scientific and practical significance. The research topic is relevant.
In order for the article to be better perceived by the readers, several improvements can be made.
It would be good to make the abstract a little more detailed. Its volume should be about one and a half times larger.
Figure 3 is not of very good quality and there is a feeling that this is a scan from a book. If this is a scan, then there should be links to the source. Also, figure 6 is missing in the article. In general, the drawings are decorated in different ways, some drawings are framed, others no. It would be good to bring everything to uniformity.
Based on the results of chemical analysis the authors found the liquation of elements along the casting section. At the same time, it is not specified from which part samples were taken for mechanical tests. It would be good to indicate this in the “Materials and Methods” section, since the properties will be different from every different chemical composition according to the casting cross-section.
In this section it is also necessary to describe in more detail the process of conducting melting – the mass of the casting, the melting time, what was deoxidized, etc.
The authors talk about general liquation due to different solidification rates, but practically do not touch on the issue of dendritic liquation. Although dendritic liquation will have an active influence on the formation of dendrite axes and on the size of dendrite axes.
Much is said about the formation of dendrite axes, but there are practically no photos of the dendritic microstructure. There is only Figure 3 and it resembles a scan from a book. It would be good to bring your own photos of the emerging dendritic microstructure with an indication of the sizes of the dendritic axes at various crystallization parameters. Also, bringing SEM photos of the dendrites of the alloy matrix with element-by-element mapping would significantly increase the scientific value of the work.
Author Response
Comments and Suggestions for Authors
In general, the study was performed at a good level. The presented results of the performed experiments have scientific and practical significance. The research topic is relevant.
In order for the article to be better perceived by the readers, several improvements can be made.
It would be good to make the abstract a little more detailed. Its volume should be about one and a half times larger.
Answer:
Thanks for the comment. Metals (MDPI) requires the abstract not to exceed 200 words. The Abstract has been improved.
Figure 3 is not of very good quality and there is a feeling that this is a scan from a book. If this is a scan, then there should be links to the source. Also, figure 6 is missing in the article. In general, the drawings are decorated in different ways, some drawings are framed, others no. It would be good to bring everything to uniformity.
Answer:
Thank you for your comment. Figure 3 is not a scan from a book! Its quality has been improved to 1000dpi. Figure 6 is on pg. 11, and the text referring to it is on pg. 11 (between lines 369 -379). All graphs and figures follow the same pattern when referring to the same result set. All Figures have been enlarged so that the legend details are better visualized.
Based on the results of chemical analysis the authors found the liquation of elements along the casting section. At the same time, it is not specified from which part samples were taken for mechanical tests. It would be good to indicate this in the “Materials and Methods” section, since the properties will be different from every different chemical composition according to the casting cross-section.
Answer:
Thanks for the comment. The sentence "Samples for tensile testing were taken every 5mm from the cooled base (metal/mold interface) of the castings, i.e., 14 specimens were sectioned for each alloy casting." was added to section 2.3 (page 4, between lines 194 and 196)
In this section it is also necessary to describe in more detail the process of conducting melting – the mass of the casting, the melting time, what was deoxidized, etc.
Answer:
Thanks for the comment.
The following sentence was inserted on pg. 6 (between the 242 and 251 lines):
"The liquid/solid transformation is accompanied by the release of thermal energy, with a moving boundary separating the two phases of distinct thermophysical properties. The analysis of heat transfer during solidification has essentially two objectives: the determination of the temperature distribution and of growth/cooling rates along the length of the castings. During the cooling of the ingots, the main modes of heat extraction are: i) thermal conduction in the metal and in the mold; ii) Newtonian transfer at the metal/mold interface; iii) forced convection of the cooling liquid at the mold bottom. Considering the position 85mm from the metal/mold interface for both cooling curves presented in Figures 2a and 2b, the time of passage of the solidus isotherm by such position in the castings were 285s and 220s, for Al-7Si-0.5Mg and Al-7Si-1Mg alloys castings, respectively."
The authors talk about general liquation due to different solidification rates, but practically do not touch on the issue of dendritic liquation. Although dendritic liquation will have an active influence on the formation of dendrite axes and on the size of dendrite axes.
Answer:
Thanks for the comment.
The following sentence was inserted on pg. 7-8 (between the 276 and 289 lines):
A process of separation diffusion occurs at the solid/liquid interface of growing dendrites giving rise to a chemical inhomogeneity from center to surface of dendrite arms. Such a phenomenon has been initially called dendrite liquation, although the term microsegregation is more commonly used in the literature [39]. The local solidification cooling rate along the length of the casting affects the composition in the solidification range, i.e., increase in cooling rate leads to progressive increase in segregation and thus in the amount of non-equilibrium eutectics [40]. Although the phenomenon of microsegregation is outside the scope of the present work, it is worth noting that the solidification thermal parameters, i.e., cooling and growth rates of the present study may have an important role in the quantification of microsegregation. As shown by Meza et al. [41] in a study on directional solidification of Al-Fe and Al-Cu alloys, the experimental microsegregation profiles from the central parts of dendrites (Al-Cu) and cellular (Al-Fe) cores were shown to vary with the growth rate from the central part of dendrite/cell cores to the limit of intercellular/interdendritic regions.
[39]. Yu. A. Samoilovich, V.I. Timoshpol`ski, S.M. Kabishov, Calculation of Dendritic Liquation (Microsegregation) in a Solidified Steel Ingot with Allowance for Ionic Migration, Journal of Engineering Physics and Thermophysics, v.82, pp. 98-105, 2009
[40]. D.G. Eskin, Structural Factors of Dendritic Segregation in Aluminum Alloys, Russian Journal of Non-Ferrous Metals, v.49, pp. 373-378, 2008
[41]. E.S. Meza, F. Bertelli, P.R. Goulart, N. Cheung, A. Garcia, The Effect of Growth Rate on Microsegregation: Experimental Investigation in Hypoeutectic Al-Fe and Al-Cu alloys Directionally Solidified, Journal of Alloys and Compounds, v. 561, pp. 193-200, 2013.
Much is said about the formation of dendrite axes, but there are practically no photos of the dendritic microstructure. There is only Figure 3 and it resembles a scan from a book. It would be good to bring your own photos of the emerging dendritic microstructure with an indication of the sizes of the dendritic axes at various crystallization parameters. Also, bringing SEM photos of the dendrites of the alloy matrix with element-by-element mapping would significantly increase the scientific value of the work.
Answer:
Thanks for the comment. Figure 7 has been enhanced and a SEM image has been inserted highlighting the essentially dendritic morphology of the Al matrix. In Figure 3, marks have been inserted highlighting the primary, secondary, and tertiary dendritic arm spacings.

Round 2
Reviewer 2 Report
The quality of the drawings has been improved. The authors answered the questions posed. In general, the article has become better.